# Implementation and Acceptance of Information and Communication Technology Incorporated into Long-Term Care

**DOI:** 10.3390/healthcare10071253

**Published:** 2022-07-05

**Authors:** Yi-Cheng Chiang, Yin-Chia Hsieh, Fan Wu

**Affiliations:** 1Department of Information Management, National Chung-Cheng University, Chia-Yi 621301, Taiwan; sofia12130406@yahoo.com.tw; 2Taichung Tzu-Chi Hospital, The Buddhist Tzu Chi Medical Foundation, Taichung 427213, Taiwan; 3Department of Business Administration, National Chung-Cheng University, Chia-Yi 621301, Taiwan; carrie.1207@hotmail.com

**Keywords:** long-term care, homecare, ICT systems, quality of medical services, long-distance care, market research technique

## Abstract

Every country in the world is facing serious demographic aging, since the average life expectancy is consistently increasing. Agencies involved in the implementation of caregiving through long-term care institutions can develop more convenient approaches using information and communication technology to enhance overall efficiency. Communication technology has enabled the strengthening of physiological instruments, improving the efficiency and quality of services, while integrating management systems for optimum efficiency. This work conducted empirical studies, collecting responses to questionnaires from residents and caregivers in five institutions located in the south of Taiwan. The PZB model, proposed by Parasuraman, Zeithaml, and Berry, was used to construct the questionnaire to analyze the service quality following the incorporation of information and communication technology. The results of the empirical study show that 34% and 63% of the relatives of the residents agreed and strongly agreed that the system was practical and convenient, respectively. As for the caregivers, 77% of them agreed or strongly agreed that the system was mobile, practical, and convenient, and they agreed that the system could significantly increase working efficiency, reduce waiting time, and improve administration for chronic diseases among care-home residents.

## 1. Introduction

Aging increases the probability of the occurrence of chronic diseases, and aging-related chronic diseases have become an important issue in public health and medical service systems worldwide [1,2]. Most developing and developed countries have been facing serious problems because of the rapid aging of their population structures. Many countries, including the Netherlands, Germany, and Japan, have also established long-term healthcare systems. To enhance the popularity and accessibility of service resources for the elderly, most governments have developed plans to establish daycare centers in townships, attracting thousands of caregivers to provide family-based and community-based services [3,4,5].

People suffering from dementia and accessing these services often require one-on-one care. Other people suffering from various chronic diseases, such as stroke, cataracts, heart disease, and diabetes, may also require a great amount of manpower to address their poor mental states. Clearly, a lack of resources in terms of manpower, funding, and organization is a barrier hindering governmental daycare policies [6,7]. Long-term care is intended to provide services that include continuing medical treatment, healthcare, nursing, nutrition, care of life, and individual and social support for people with physical and mental dysfunctions, dementia and disability for a certain period. To avoid the elderly being housebound and lacking sufficient privacy and independence in their lifestyles, many countries have widely promoted the concept of aging in place through social welfare [8].

With the rapid advancement of science and technology, many breakthroughs have been achieved in this field as well as in the study of certain chronic diseases, thereby promoting the further development of health. Long-term care and aging in place can be further combined with the latest information and communication technology to successfully allow the elderly to deal with aging efficiently and effectively [3,9,10,11]. More importantly, the development of information and communication technology can overcome some of the limitations of aging in place, thereby helping the elderly to live at home [12,13,14]. Consequently, labor demands are reduced, and resources can be allotted to further promoting care activities.

In general, medical ICT involves monitoring vital signs at home for patients with chronic diseases, to understand their physiological conditions, and utilizing video calls in lieu of nursing home visits [15,16]. Remote homecare systems for the self-management of patients with chronic diseases include the monitoring of clinical signs, the automation of dispatch, and reminders for medication; in this way, health education and daily records are conducted at the same time [17,18]. These endeavors were established using information and communication technology to make healthcare management more convenient [18,19,20]. Furthermore, many clinical case studies have shown that the early detection of adverse clinical phenomena through self-monitoring records taken at home significantly reduces readmission rates and hospital stays [21].

Figure 1 shows the infrastructure constructed for the long-term care society. The home of the elderly individual can be equipped with sensors such as vital-sign rings or other equipment, such as sensor mattresses, to detect his/her sleeping status, and even indoor locating equipment, which can collect the health-status behavior and movement of the individual through multiple triangulation by measuring the relative strength of ZigBee signals. The data are collected and sent to the long-term care platform, which is typically established by the government. Then, the data, in turn, are stored in a cloud database for further monitoring by a call center; the relatives of the elderly patients and the caregiver can also obtain the data by contacting the authorities. The ICT equipment should facilitate manual or automatic emergency calls, to reduce the time gap in helping/curing the patients.

This study constructed a module of information and communication technology by implementing the transmission of medical instructions through 4G and/or Wi-Fi. With the wireless infrastructure and cloud operations, service quality and efficiency can be enhanced. The achievements of this work are as follows:The investigation of utilizing information and communication technology for increasing service efficiency and quality and constructing a new, efficient, and precise form of care.The installation of biometric instruments for health enhancement, chronic disease management, and safety surveillance, monitoring blood pressure, temperature, pulse, etc. The data can be uploaded to the cloud such that relatives can monitor the biometric values of the residents anytime and anywhere, and interactive cloud healthcare management can be established.The use of the modified service agreement model PZB [22,23,24] to conduct a questionnaire for residents, residents’ relatives, and caregivers about heath monitoring, chronic disease, and drug safety before/after the incorporation of the system.

### The Literature

With the ever-developing techniques and products, whether the techniques and products are accepted or rejected by users is worth investigating. In the past decade, many studies have explored users’ acceptance of new products/techniques and quantitatively evaluated their favoritism, intentions, and loyalty [25,26]. The technique acceptance model (TAM) and theory of reasoned action (TRA) [25] utilize the psychological theory of reasoned action and theory of planned behavior to understand and explain human behavior regarding the potential acceptance or potential rejection of new techniques and products. Those acceptance models are crucial, but they place less importance on certain aspects. Customer needs not only relate to the techniques and products, but also the services associated with the products. The service quality relates to the added value to the customers as well as the value chain. The service quality gap model (also known as the PZB model) was developed by three major researchers, Parasuraman, Zeithaml, and Berry [22]. They believed that quality was of the highest priority regarding the expectations for services of customers. PZB model [22] proposed insights and propositions from consumer’s viewpoint, revealing 10 dimensions (or indicators), namely, reliability, responsiveness, competence, access, courtesy, communication, credibility, security, understanding, and tangibles, that influence the consumers’ expectations about the service.

According to the concept of the service model (called the PZB model), a model based on that model was used to design the questionnaire for service quality [22,23,27]. Wang et al. [23] collected contributed papers published from 1998 to 2013 that used “SERVQUAL (Service Quality)” model to gauge service quality, and gave a comprehensive survey about the trend and contribution of the service quality. In [27], Ko et al. also used SERVQUAL as well as importance performance analysis to identify the dissatisfactory factors in nursing home. The literature shows the service quality should be well gauged to find the potential problems.

In the PZB model, there are five insufficiencies that reduce satisfaction with service provision and propagation. Enterprises should address the five insufficiencies to satisfy customers’ requirements [28]. The five insufficiencies were constructed as five aspects in the questionnaire: morphism, reliability, responsiveness, assurance, and empathy. The five insufficiencies of the PZB model are as follows [22]:Insufficiency of service knowledge: the gap between the customer’s expectations and the manager’s understanding of such; the manager is not aware of the service required by the patients.Insufficiency of service standards: the gap between the service actually provided by the manager and the expectations of the customer; for example, the managers translate their idea of the required service into a certain standard of service, but the standard of service received by the customers falls short.Insufficiency of service provision: the gap between the ideal service specification and the actual service, which is influenced by the training received by the service providers.Insufficiency of service communication: the gap between the actual service provision and the service implied by the external communication from the service provider; the advertisements form the external communication between the service provider and customer. The service expectations of the patient or population will be formed by the external communication. The insufficiency of service communication is based on the above gap.Insufficiency of cognition: the gap of cognition in expected service and actual service, for example, the patient and the population have a cognition gap between the expectation and the received service.

Healthcare involves professional-oriented and service-oriented activities. The customer (i.e., patient)’s satisfaction is the major concern of the healthcare provider and patients. The TAM and TRA cannot fully describe the service expectation gaps for techniques and products. Thus, this study adopted the PZB model as the major theoretical basis for assessing the acceptance of a long-term care system using ICT techniques. Previous research [25] adopted the PZB model to assess the incorporation of tele-healthcare, while other research [26] used the PZB model to evaluate the expectations and satisfaction of hospitals. Lim and Tang [26] utilized the market research techniques, i.e., SERVQUAL, to determine the expectation and perception of patients, and finally suggested six improvements, including tangibility, reliability, responsiveness, assurance, empathy and accessibility and affordability, to the hospitals. For the PZB model used for medical services in this study, its description is as follows:Morphism: medical institutions have modern equipment and pay attention to staff outlook.Empathy: medical institutions sympathize with patients regarding their ailments and consider the improvement of their health as the first priority.Reliability: medical institutions can provide appropriate and adequate services to ensure that the patients feel that they are reliable.Responsiveness: medical institutions can actively solve the problems of the patients and provide efficient medical services.Assurance: the medical staff all have good discipline in their professions.

## 2. Materials and Methods

The daycare center mainly provides services for people with mild dementia, those with disabilities or chronic diseases, and people who are in stable healthy conditions but require continuous care after leaving hospital, as well as patients with limited or good mobility, or people who lack the ability to deal with their daily lives. Most of the protocols for caring for the elderly prefer the elderly to be living in their own homes or familiar communities. However, if the elderly want to enable their children, who would care for them, to work easily, they still can obtain appropriate entrustment and care when they choose daycare services that provide professional nursing services to meet their needs during the day and then enjoy family life during the night. The daycare service—which includes medical care, nursing, rehabilitation, healthcare, nutrition services, and care for elderly—allows them to enjoy their social life with dignity and humanity. It is a reasonable assumption that the relatives care or even worry about health status when their elderly relatives stay in daycare centers. The dynamic and real-time information of the elderly patients and their aggregated health data are expected to be remotely accessible for the relatives.

The medical equipment used in long-term care facilities includes electronic blood pressure gauges, medical electric scooters, thermometers, blood glucose meters, and other important products from international suppliers [12,29]. The installation of ICT systems for long-term care in the community offers many advantages, including reminders and foolproof functions, the prevention of human error, the rapid completion of work and reduction of transcribing work, significant enhancement of the quality of care and efficiency, and an increase in administrative capacity. Meanwhile, it can introduce a standard operating system with professional capabilities and accumulated experience, facilitating the collection and organization of evaluation data. In addition, the burden on caregivers can be reduced, thus providing more time for other individual cases, resulting in better physical and mental healthcare for each case [30,31,32].

Figure 2 shows the whole system, which was constructed by using a Microsoft suite solution and run on a PC server. The PC server, shown in the middle part of Figure 2, also called the backend in this paper, functions as a data center, and it adopts the Microsoft internet information server (IIS) as the gateway by which to collect and transmit data from and to the Internet. An application written in C# is run behind this IIS to handle the reply to a query and store the input data in the database (using MS SQL Server). A barcode reader is used to identify the residents and users themselves before using the instrument to measure residents’ temperatures, pulses, and respiration (TPRs). The processing of the measured TPRs is shown in the left part of Figure 2, and this is called the frontend of the system. The caregivers, managers, and relatives can browse the residents’ health data remotely after completing password validation. The remote query environment is called the daycare end system, shown in the right part of Figure 2. The system collects the physiological information from the frontend sensor node and transmits it to the backend database and website service platform, providing a common platform for health services integrated from the sensing end to management end with professional analysis and consulting.

To integrate the network architecture for health, the frontend physiological signal monitoring system can be transmitted over Bluetooth and Wi-Fi through cellphone and notebook intermediary service platforms to integrate the processing of physiological information acquired from the frontend sensing signal. The captured physiological signal is transmitted over the service platform of the network system to the database in a healthcare center for storage. As for backend analytical processing, judgments and emergency notifications are referred to the healthcare institute and hospital professionals for analysis and judgment, which can reduce the reporting work for caregivers. Human resources can be utilized efficiently by applying the scale to evaluate chronic cases of mental and physical conditions. Afterward, the outcomes and results of analysis are transmitted via the healthcare service to the network service platform, which can then perform any required procedures (e.g., recommend an exchange between physicians and caregivers, and notify the family). 

The incorporated system integrates healthcare, information, and communication to a cloud service. With the introduction of the system, the daycare center can use RFID for identification and ZegBee for transmission to automatically collect biometric data into the system [31]. Thus, the daycare center does not need caregivers to manually enter data; the costs are reduced and the income increases. The daycare center system consists of five system modules, which are as follows:

1. Daycare administration module:

Through the administration module, the staff can process the admission and discharge of residents, including the resident’s demographic data, emergency contact details, relative data, off-applying, financial data, bed data, personnel data, etc. The staff can observe the distribution of the occupied spaces, the visiting of relatives, and comments from patients.

2. Healthcare module:

Through the healthcare module, the staff can easily enter the data of evaluations, nursing care plans, nursing shift records, periodical health exams, and laboratory tests, which form chronological records. The staff and the relatives can inquire about the current and past health statuses of the residents, which can be indicators for diets, medicine, and healthcare. This module stores the healthcare data in the system, which can integrate the daycare administration module, such that the alarm and reminding mechanism can be implemented. Therefore, the drug safety and healthcare monitoring for residents can be enhanced, the manual transcription of records can be reduced, and, finally, appropriate healthcare services can be provided.

3. Social-worker module

Through the social-worker module, the staff can provide an automatic system for social workers such that they can computerize the plans for their social work, arrange social resources, and administrate social group activities and referral records. The module can be further integrated with the two modules of healthcare and administration such that the staff can comprehensively query the statuses of residents online.

4. Decision-support module

Through the module, the staff, administrator, and state holders can easily trace exceptional events and process them. The events include clinical cases, such as adverse drug reactions, infections, and falls, and administration cases, such as being over-budget and low utilization rates. With the integration of all the modules, the administrators and stakeholders can track indicators for facilitating clinical and administrative planning and decision support.

5. Web-based-service module

With this module, the daycare center can keep up with the Government’s broadband plan, which asks all institutions to provide web-based services to provide remote access for relatives. The module enables bi-directional interactive video conferences, such as web conferences, for relatives, so they can find out the health statuses of the residents and leave messages for the residents and institutions. With the module, the institution can construct a sincere, responsible, and kind work process for the residents and their relatives.

The system provides the instruments with which to measure blood pressure (arm-based measurement), pulse, and temperature. Ear-temperature measurements are transmitted through Bluetooth; then, the data can be propagated to the system without the intervention of staff. The flowchart for the measurements is shown in Figure 2.

It is worth noting that, for each measurement by the electric sphygmomanometer and ear thermometer, the data are first confirmed by the caregivers before their propagation to the designed system. The demographic data of the residents are established in the system beforehand, such that the measurement results can be connected with their individual databases; furthermore, the system can e-mail the vital-sign data to the relatives such that the residents, institution, and relatives can remain up to date regarding the vital-sign status.

According to research of Wolak [32], the service can be considered non-morphological, heterogeneous, indivisible, and perishable; other scholars also considered that service quality should meet the expectations of the customers [33]. The consumers use their feelings regarding the service received and service process to evaluate the service quality; obviously, the perceived service quality is based on the subjective cognition of the consumers. With respect to the service quality in a healthcare institution, the quality of the healthcare services directly influences the overall service quality of the hospital. Furthermore, for hospitals with similar healthcare technologies, the healthcare service quality impacts the degree of satisfaction of populations with the hospitals.

## 3. Results and Discussion

From the long-term healthcare point of view, the healthcare system established by using the ICT would not be easily built up and might take a long time to establish. In addition, this system also requires support from the heads of medical organizations and community units, from medical staff (i.e., caregivers), and from residents and their relatives. Therefore, in addition to considering the needs of residents and their relatives in designing the ICT, caregivers should be taken into account in terms of the convenience and efficiency.

To evaluate the effectiveness of the incorporation of an ICT system into a daycare center, a questionnaire based on the theory of the PZB model was used to determine the levels of acceptance of the users and patients. Since the system is implemented in a daycare center, all the residents are in sub-healthy conditions. Most of the elderly individuals are able to walk, but slowly, and are in reasonable health. Some of the residents have serious health conditions, such as paralysis since strokes and dementia; some are solitary but require daycare. Since the individuals would not use the ICT devices and system, the questionnaire was mostly answered by the relatives, but there were still some residents who could answer the questionnaire by themselves. The PZB is a multi-dimensional instrument designed for evaluating user expectations and perceptions of services [22,23,24]. Five dimensions—tangibles, reliability, responsiveness, assurance, and empathy—form the SERVQUAL measure and are used to characterize service quality [34]. This study adopted the theory and designed a questionnaire for the daycare center for evaluating the incorporation of the system. Since two types of users are involved in the system—one being the caregivers and the other being the residents—the questionnaire had two sets of responses for the above two types of users. The contents of the questionnaires were classified into five parts, according to five dimensions of service quality, as follows:

1. Tangibles:

The questions intended to evaluate the practicality and convenience of the instruments used in the system; moreover, the questions intended to evaluate the degree of satisfaction of the relatives of the residents with the system and biometric data collections, and the degree of satisfaction of the caregivers with the system and the efficiency of the care provided through the system.

2. Reliability:

The questions intended to evaluate the time saved in the collection of biometric data and effectiveness of disease control and enquiries into healthcare status, in addition to the reduction in the workload of the caregivers.

3. Responsiveness:

The questions intended to evaluate the financial and care burdens on the residents’ relatives and the feelings of the relatives. In addition, the questions also intended to evaluate the working efficiency and flaws of the caregivers.

4. Assurance:

The questions intended to evaluate the relationships of the relatives of the residents and caregivers and the degree of endeavor in the care.

5. Empathy:

The questions intended to evaluate the requirements of the relatives of the residents and the feelings and attitudes of the relatives and caregivers regarding the system and instruments, including the continuums and recommendations.

IBM SPSS 20.0 was used to analyze the questionnaire responses. The questionnaires were used to interview 210 residents or their relatives and 28 caregivers. Most of these 210 persons who answered the questionnaire were relatives of the residents, whose demographic data can be seen in Table 1; we have used the term “relatives” to denote all these 210 persons for brevity. The answers to the questionnaires were collected from five institutions in the south of Taiwan that incorporated the ICT system. Since the incorporation of the system provides fast responses in the measurement, querying, and evaluation of health data, most of the relatives and caregivers were satisfied with the benefits obtained from the system. Thus, most of the relatives were glad to participate in the survey and respond to the questionnaire. In the descriptive statistics, the continuous variables are reported as the mean plus variance; the categorical variables are reported by frequency. The *t*-test was used to evaluate the differences between before and after incorporating the information and communication technologies in the institution. Furthermore, factor analysis was used to construct the questionnaire, determining five aspects to explain and name the mutual relationships of the variables in the questionnaire. Using a multiple-regression model, the service quality was analyzed with respect to the demographic variables of the residents and their relatives, and then, the related models were constructed. The criterion of *p* < 0.05 was adopted as indicating a significant difference in the analysis. The demographic data for the residents show that, of the 210 residents, 56 were males and 154 were females; i.e., 73.3% were female. In terms of the age distribution, those aged 60 or above formed the largest group, while those aged 51–60 were the second-largest group. In terms of education levels, those educated to university and college level formed the largest group, accounting for 59.5%, while the second-largest group was those educated to high school level, accounting for 27.1%. In terms of the lengths of the stays of the residents, those who stayed for over two years were the most numerous, accounting for 41.4%; the second-largest group was those staying for one to two years, accounting for 38%. Regarding the distribution of the residents by cognitive status, 35 cases had a slight degree of dementia, accounting for 23.3%; 20 cases had a medium level; and 20 cases had a heavy–medium level, but none showed heavy levels (except as reported in the questionnaire).

In investigating the distribution of the service quality of the system as reported by the 210 relatives of the residents based upon the morphism of the PZB service model, 63% of the relatives strongly agreed and 32.7% agreed with the introduction of the system capable of providing practicality and convenience for the staff from the viewpoint of the relatives. Regarding the reliability of the service model, 56% of the relatives strongly agreed with the idea that the system could reduce the operating time in recording the biometric data, 49.3% of the relatives thought that each measure could reduce it by more than 2 min, and 23.3% thought that each measure could be reduced from 2 min to 30 s. The relatives also thought that the control of blood pressure and health status could be made more stable and real time through the utilization of the system; 72.4% of the relatives strongly agreed and 16% agreed with the above idea about the control of blood pressure and health status. Regarding the responsiveness of the service model, 62.7% of the relatives strongly agreed and 27% agreed that the incorporation of the system could reduce the financial burden of managing chronic diseases and reduce the care burden. Regarding the assurance with the service model, 42.7% of the relatives strongly agreed and 39.5% agreed that the relationship between the relatives and caregivers could become closer, and 42.6% strongly agreed and 22.4% agreed that they could have more time to accompany the residents. Regarding the empathy with the service model, 50.4% of the relatives disagreed with and 13.9% had no comment about the idea that the requests and feelings of the residents could be looked up before the introduction of the system, but this changed after the introduction; then, 48% of the relatives strongly agreed and 31.5% agreed. Regarding the loyalty with the service model, 68% of the relatives strongly agreed and 17.2% agreed that they would recommend the system to others. Regarding the satisfaction with the service model, 66.2% of the relatives strongly agreed and 14% agreed that they were satisfied with the monitoring of blood pressure. In addition, 36.6% of the relatives disagreed with and 41.7% had no comment about the service degree of the daycare center, but this changed after the introduction; up to 79% of the relatives then strongly agreed. Similarly, as for the awareness of the physical conditions of the residents, 61.6% of the relatives disagreed with the issue of awareness, but this changed after the introduction; then, up to 82% of the relatives strongly agreed with the awareness issue. Finally, the difference between the satisfaction before and after the introduction of the system was also statistically significant; the *p* value was 0.00001.

Questionnaires were also conducted for the caregivers. All the caregivers were female, aged 35 to 58, and well-trained in the medical care fields. Investigating the distribution of the service quality of the system as reported by the caregivers showed that 75% of them strongly agreed and agreed with the introduction of the system capable of providing mobility, and 62.5% of them strongly agreed and agreed that it was practical and convenient. Regarding the reliability of the service model, 62.5% of the caregivers strongly agreed and 37.5% agreed that the system could reduce the operating time, increase the work efficiency, and reduce the potential errors in recording the biometric data. Regarding the management of cases of chronic disease, 62.5% of the caregivers strongly agreed and 37.5% agreed that this system could speed up operations; in detail, the time spent on recording could be reduced by 60 to 90 s per case. Regarding the guarantees of the system, 62.5% of the caregivers strongly agreed and 32.5% agreed that the system could decrease the distance between the relatives and caregivers. Regarding empathy, the caregivers could be more considerate of the requirements of the cases. Of the caregivers, 69% were willing to continuously use the system and would recommend it to other institutions. After the usage of the system, 62.5% and 37.5% of the caregivers agreed that they were satisfied with the measurement of blood pressure, the reduction in recording time, and the increase in efficiency in measuring the biometric data. The difference in the overall satisfaction with the system before and after the usage of the system was significant according to the criterion *p*; *p* = 0.00001.

From the results of the above questionnaire for two types of users (the residents or their relatives and caregivers), we can observe that there are slight differences in the assessment of the ICT system according to the different roles. Up 95.7% (63% strongly agree plus 32.7% agree) of the relatives agreed that the system was practical and convenient for the staff, while only 62.5% of the caregivers agreed. A possible reason is that the relative can query the healthcare data remotely, but the caregivers need to install, connect, and push some unfamiliar buttons when they perform data acquisition on the gadget. Learning how to use extra equipment is a burden for them. This would imply that about half of the relatives (49.3%) thought that the measurement of health data was made easier, with the measurement time being reduced by up to 2 min, while only around one-third of the caregivers (i.e., 37.5%) agreed that the time was reduced, and they believed that the reduction was 90 s rather than 120 s. Regarding the communication between the relatives and caregivers, both types of user (81% of the relatives and 94.5% of the caregivers) agreed that the communication lag and distance were significantly decreased. It can be speculated that, if the data can be queried by the relatives online, any abnormal or exceptional conditions can be more easily conveyed by the system. The recommendations appear to be the most important criterion for the system; 85.2% (i.e., 68% strongly agreed + 17.2% agreed) of the relatives were happy to recommend the system to others if they had their families living in the long-term care institution, while 69% of the caregivers would continuously use the system and recommend it to other institutions. From the above analysis of the questionnaire responses for the two types of users, we can conclude that the ICT system, if incorporated into a long-term care institution, can be beneficial to both caregivers and relatives.

Since the results were obtained from only five institutions in the south of Taiwan after their incorporation of the system, the results have several limitations. First, the previous healthcare system of the institutions used by the caregivers will influence the adoption of the system. The greater the degree of automation in the previous healthcare system, the greater the degree of adoption by the users. Secondly, the educational levels of the relatives and elderly individuals will influence the acceptance of the system. It is believed that the higher the educational levels of the relatives and elderly patients, the greater the satisfaction of the relatives and patients. Thus, the results of the research are not directly applicable to other institutions if their scales and investment are different. Due to these limitations, the research results may not be generalizable to all long-term care institutions. However, the results can be considered as a feasibility assessment and acceptance evaluation of the information and communication system’s incorporation into long-term care institutions. The management of future healthcare models should be sustainable and complete, which is the demand and opportunity of the market in the healthcare industry. With the PZB-evaluation results of the study, we confirm that the duplication and dissemination of the ICT system are feasible and that similar healthcare industries can introduce the ICT system in response to the needs of healthcare. That is, healthcare institutes can install monitoring mechanisms to control personnel access; control the use of drugs, equipment, and facilities; strengthen security systems for protection and prevention; reduce the costs of healthcare centers; and enhance the quality of healthcare services.

## 4. Conclusions and Future Research Works

The rapid aging of the population due to the increase in average life expectancy and the decline in birth rates is causing many problems in society, such as with insurance and national economics. Healthcare for the elderly includes medicine, nursing, social work, occupational therapy, physical therapy, nutrition, and other services, and professional capabilities are required in the different areas. At present, homecare, daycare centers, and other long-term care facilities all face the problem of unaffordable manpower. The development of information and communication technology (ICT) could create the opportunity to improve convenience for caregivers. ICT associated with physiological instruments, and computerized management systems can enable stable and near-optimal care, as well as shortening processing times and increasing the safety of residents. The introduction of ICT can also provide significant convenience and effectiveness for staff, enhancing the management of residents’ chronic diseases and the mental health of care personnel, and enabling the instant recall of the latest healthcare records of the residents. Since network speeds and computing power have significantly advanced in recent years, we recommend the early adoption of the ICT system by long-term caregivers. The ICT system can integrate various systems and facilitate the implementation of healthcare policy with minimal effort.

This study adopted the PZB model to conduct the survey of service quality for the introduction of the system, focusing on the attitudes of caregivers and residents. The questionnaire gathered information to compare monitoring, the management of chronic disease, and medicine before and after the incorporation of the system. The results show that, from the perspectives of both the residents and caregivers, the efficiency and convenience of working were significantly increased and the workload was dramatically reduced. In addition, following the introduction of the system, the satisfaction of the relatives with the daycare center significantly increased, and they, in turn, were happy to recommend the system to other daycare centers. Future work will incorporate more and newer information technologies to long-term care institutions, including big data techniques to perform health data prediction, trend analysis, and the verification and validation of health data, and artificial intelligence techniques to realize warnings and reminders for healthcare operations based on the current system, so as to reduce caregivers’ burdens and simultaneously increase the quality of healthcare.

## Figures and Tables

**Figure 1 healthcare-10-01253-f001:**
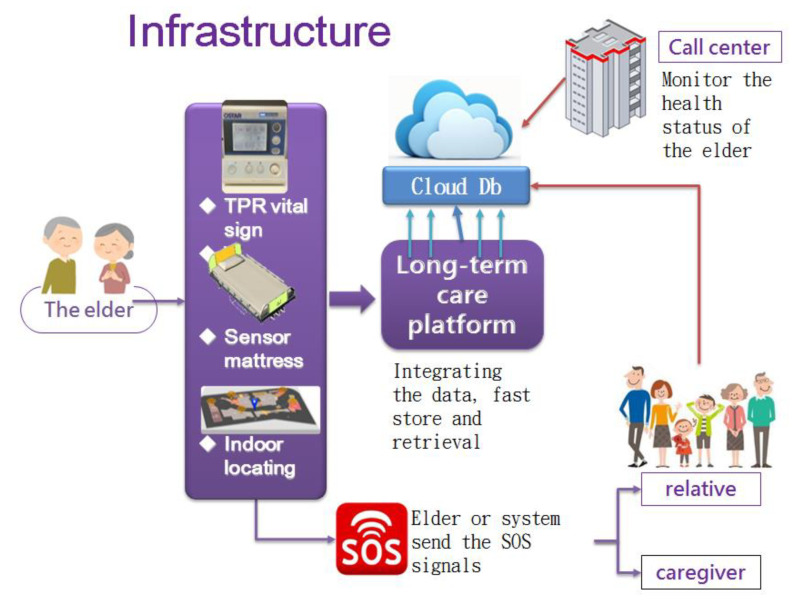
The infrastructure for the incorporation of ICT into long-term care (note: TPR means temperature, pulse, and respiration).

**Figure 2 healthcare-10-01253-f002:**
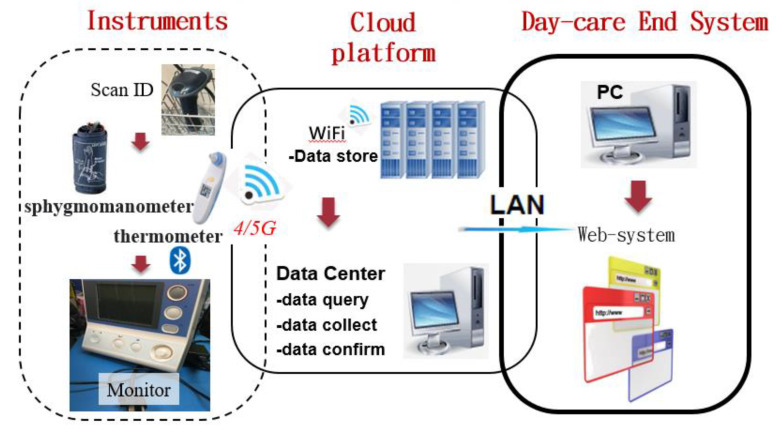
The transmission of biometric data.

**Table 1 healthcare-10-01253-t001:** Demographic profile of the residents.

Demographic Data (=210)	*n*	%
Gender		
Male	56	26.7
Female	154	73.3
Age		
50 years old or less	7	3.3
51 to 60 years	52	24.7
61 years old or more	151	71.9
Education level		
University and college	125	59.5
High school	57	27.1
Low to high school	28	13.3
Length of stay		
6–12 months	42	20.0
12–24 months	81	38.6
24 months and up	87	41.4
Cognition status		
Slight	96	45.7
Medium	70	33.3
Heavy–medium	44	20.9

## Data Availability

Not applicable.

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
