# Peer review of "Implementation and Acceptance of Information and Communication Technology Incorporated into Long-Term Care"

_healthcare, 2022, doi:10.3390/healthcare10071253_

Round 1
Reviewer 1 Report
Review on healthcare-1740420
Implementation and Acceptance of information and communication technology incorporated to long-term care
This paper has generally well written, however, in some parts, there are serious concerns. All in all, it has some merits, however, there are some comments as follows which should be addressed so as to strengthen the quality of the manuscript:
- It is strongly suggested to separate Introduction and Literature review sections.
- The literature should be strengthened, specifically with the relevant and recently published papers.
- The main contributions of the manuscript should be clearly stated at the end of Section 1.
- The assumptions should be clearly stated at the beginning of Section 2, and before stating the problem on-hand.
- The characteristics of the used PC, and all other software should be clearly specified at the beginning of Section 3.
- How did the authors generate the sample instances?
- How did the authors tune the used parameters, reported in Table 1? Through trial and error, benchmarking or other references? Please clarify it.
- Some managerial and practical implications should be extracted according to the obtained results, and analyses.
- The conclusion section must be improved so that more details about the outcomes, and conducted analyses are included.
Author Response
For the 1st reviewer comments:
Comment 1: It is strongly suggested to separate Introduction and Literature review sections.
Reply 1: Thanks for the comments, which will be more clear for the readers. We have separate the introduction and literal review into two sections, and add several newly published and related paper in this version.
Comment 2: The literature should be strengthened, specifically with the relevant and recently published papers.
Reply 2: Since this revised version has separate the literature as a new section, we have added more papers to enhance the survey related to our paper (some of which are published in this journal, in this version).
Comment 3: The main contributions of the manuscript should be clearly stated at the end of Section 1.
Reply 3: Thanks for the reminding. We have revised the sentences to clearly state our contribution in the end of section 1.
Comment 4: The assumptions should be clearly stated at the beginning of Section 2, and before stating the problem on-hand.
Reply 4: Thanks for the comments. In this version, we add a paragraph to talk about the role of daycare center in modern society and why the relatives need the ICT system to let them to access the real-time and aggregated health data of their elders at the beginning of Section 2. We believed some elders go to the daycare center to get continuous care after leaving hospital, and some elders want to help their children work at ease. The elders select daycare services that provides professional nursing service to meet the needs of the elderly during the day and enjoy family life during the night. It is a reasonable assumption that the relatives care or even worry about the health status when their elders stay in daycare center. The active and in-time information of the elders and their aggregated health data is expected to be remotely accessed by the relatives.
Comment 5: The characteristics of the used PC, and all other software should be clearly specified at the beginning of Section 3.
Reply 5: For clearly describing the structure and associated operations in the system, we add a paragraph in the beginning of Section 3 to describe the system’s software and hardware structure in a systematic way in this version in the beginning of Section 3.
Comment 6: How did the authors generate the sample instances?
Reply 6: We are sorry for ‘sample’ is a typo. The questionnaires are interviewed with 210 residents or their relatives and 28 caregivers, rather than sampling. The questionnaires are collected from five institutions in the south of Taiwan that incorporated the ICT system. Since the incorporation of the system provides fast response in measuring, querying and evaluation of the health data, most of the relatives and caregivers satisfy the benefits got from the system. Thus, most relatives are glad to participate the survey and response the questionnaires.
Comment 7: How did the authors tune the used parameters, reported in Table 1? Through trial and error, benchmarking or other references? Please clarify it.
Reply 7: (Continuing comment 6) we are sorry for term ‘sample’ being a typo. The questionnaires are interviewed rather than sampling. Most relatives are glad to participate the survey and response the questionnaires, no criterion to sample the questionnaires.
Comment 8: Some managerial and practical implications should be extracted according to the obtained results, and analyses.
Reply 8: We have added two paragraphs in discussion section: one is to discuss and analyze the limitation of the PZB-evaluation results. The other is to confirm the duplication and dissemination of ICT system are feasible since of the PZB-evaluation, which is the demand and opportunity of the market in the healthcare industry.
Comment 9: The conclusion section must be improved so that more details about the outcomes, and conducted analyses are included.
Reply 9: We have revised the paper as the suggestion, adding the limitation of the research results, the benefit in managerial viewpoint after its incorporation, and the future work of the research, including the incorporation of the big data, artificial intelligence techniques to reduce the caregivers’ loading and increase the quality of the healthcare.
Reviewer 2 Report
Dear Authors,
The paper conducts empirical studies, collecting 210 residents in the institution, utilizing PZB model, proposed by Parasuraman, Zeithaml and Berry, for constructing the questionnaires to analyze the service quality for the incorporation of information, and communication and technology.
This paper needs major revision.The authors need to address the following points.
1) Add Related Work section after introduction. Cite all important papers related to the topics.Therefore, after Introduction, you must add Related Work.
2) Write an detailed proposed algorithm.
3) Experimental outcome is in-sufficient, please incorporate more method to justify the experimental part, atleast two to three pages you need to add on that.
4) Result part is insufficient, please add more results with the proposed models.
5) Add another paragraph naming as Discussion and discuss analytically the outcomes of your proposed method.
6)In the conclusion part enhance the future work.
Author Response
For the 2nd reviewer comments:
Comment 1: Add Related Work section after introduction. Cite all important papers related to the topics. Therefore, after Introduction, you must add Related Work.
Reply 1: This revised version has separate the literature as a new section, and adds more papers to enhance the survey related to our paper (some of which are published in this journal, in this version). Thanks for the reminding
Comment 2: Write a detailed proposed algorithm.
Reply 2: Since the ICT system is used to collect the biometric data into the system with mobile devices and provides a web-based interface to users for querying, the data flow is simple and direct. We added a new paragraph to describe the data flow of biometric data to explain the measuring, collection, and querying processes in the beginning of section 2. The algorithm for the workflow is omitted since it is too straight.
Comment 3: Experimental outcome is in-sufficient, please incorporate more method to justify the experimental part, at least two to three pages you need to add on that.
Reply 3: Thanks for comments. In this version, we carefully compare the two experimental results from the two types of users. The comparing issues includes whether the system can provide the convenience and practice, whether the system can reduce the (psychological) distance between the caregivers and the relatives, and whether they will recommend the system to others. All the issue got the positive reply from them, which in turn show the helpfulness of the system.
Comment 4: Result part is insufficient, please add more results with the proposed models.
Reply 4: Please see the reply of comment 3.
Comment 5: Add another paragraph naming as Discussion and discuss analytically the outcomes of your proposed method.
Reply 5: We have added two paragraphs in discussion section: one is to discuss and analyze the limitation of the PZB-evaluation results. The other is to confirm the duplication and dissemination of ICT system are feasible since of the PZB-evaluation.
Comment 6: In the conclusion part enhance the future work.
Reply 6: We have revised the paper as the suggestion, describing the benefits of the incorporation of the system. The future work of the research is also added in this version, including the incorporation of the big data, artificial intelligence techniques to reduce the caregivers’ loading and increase the quality of the healthcare.
Reviewer 3 Report
This study is a paper that analyzed the usefulness of ICT for long-term observation and nursing of patients, including the elderly, through a survey. Through the opinions of 210 surveyors, it was confirmed that ICT provides various assistance in various patient nursing.
This paper analyzed the importance of ICT in future care as it was difficult to find caregivers due to the rapid increase in the elderly population along with the development of ICT.
This study is very timely and also includes global issues, but in order to increase the completeness of the paper, the followings are to be modified.
-Explain the reason for choosing the PZB model in this study in more detail and describe the other models.
- Describe whether the survey participants agreed to participate in the survey and whether they responded sincerely.
-Chapter 3's analysis is very superficial, so it is recommended to present a more in-depth analysis (e.g., correlation analysis).
- Also, describe the implication of the survey results in detail at the end of Chapter 3.
-Change the title of Chapter 4 to 'Conclusion and Future Research Works' and describe the major future research issues
Author Response
For the 3rd reviewer comments:
Comment 1: Explain the reason for choosing the PZB model in this study in more detail and describe the other models.
Reply 1: Thanks for the comments. We have added more papers to enhance the survey related to our paper, please see page 10 in this version.
Comment 2: Describe whether the survey participants agreed to participate in the survey and whether they responded sincerely.
Reply 2: Since the incorporation of the system provides a fast response in measuring, querying and evaluation of the health data, the participants, most of the relatives and caregivers, all give the good responses to the system so as to agree to participate the survey and response questionnaire sincerely. The reason why they are glad to participate the survey has been added in this version.
Comment 3: Chapter 3's analysis is very superficial, so it is recommended to present a more in-depth analysis (e.g., correlation analysis).
Reply 3: In this version, we have added the reply of the caregivers and get the similar positive altitude of the system.
Comment 4: Also, describe the implication of the survey results in detail at the end of Chapter 3.
Reply 4: We have revised the paper as the suggestion, adding the benefit in managerial viewpoint after its incorporation, the limitation of the PZB-evaluation our results, and the confirmation of feasibility if the duplication and dissemination of ICT system to similar healthcare institution.
Comment 5: Change the title of Chapter 4 to 'Conclusion and Future Research Works' and describe the major future research issues
Reply 5: Thanks for the suggestion. We have revised the paper as the suggestion, describing the benefits of the incorporation of the system. The future work of the research is also added in this version, including the incorporation of the big data, artificial intelligence techniques to reduce the caregivers’ loading and increase the quality of the healthcare.
Reviewer 4 Report
This work presents an empirical PZB-model-based study regarding the use of ICT in long-term care. Due to the interest of the topic addressed, I find the work of utility for the scientific community. However, I think that it could be suitable for publication in the Healthcare journal provided that the following comments are implemented within the document:
- It should be clearly stated in the Introduction section if previous PZB-based works have been already published regarding the use of ICT in long-term care (please include such references if appropriate). If so, which were their results? which is the innovative contribution of the present paper as compared to those? If this is the first study which analyzes such issue through PZB, then please indicate it.
- The institution from which data are collected should be specified both in the Abstract and in the Introduction section.
- The acronym "TPR" present in Fig. 1 should be introduced before its use.
- I miss in Fig. 1 a communication layer which includes ZigBee, Bluetooth, NFC, WiFi, 5G. etc.
- Results disaggregated by the variable "resident", "relative" or "care giver" should be included.
- When describing data from the residents, information regarding their current health status would be welcome (very good, good, regular or bad).
- Some data relative to the care givers should be mentioned.
- The limitations of the study should be added.
- The whole manuscript should be revised by a professional proofreading service.
Author Response
For the 4th reviewer comments:
Comment 1: It should be clearly stated in the Introduction section if previous PZB-based works have been already published regarding the use of ICT in long-term care (please include such references if appropriate). If so, which were their results? which is the innovative contribution of the present paper as compared to those? If this is the first study which analyzes such issue through PZB, then please indicate it.
Reply 1: The previous study, like TAM and TRA models for the system evaluation in psychological view has been added in this version. The previous work adopting PZB for tele-healthcare, etc., has also included. Their results demonstrate the PZB can be used to show the satisfaction evaluation of the new technologies in healthcare industry. The main difference is the paper applied the PZB model to long-term care institution who incorporated the ICT system in their operations. To the best of our knowledge, few paper made the same research using the PZB model to the Long-term care institution. The results can be a reference when they consider or implement the ICT system to their institution.
Comment 2: The institution from which data are collected should be specified both in the Abstract and in the Introduction section.
Reply 2: Thanks for the reminding. The location (the south of Taiwan) of the institutions has been mentioned in the abstract section and materials and method section in this version. Rather than mentioned in introduction section as the comment, we mentioned the location in materials and method section since the information is needed in describing the questionnaire.
Comment 3: The acronym "TPR" present in Fig. 1 should be introduced before its use.
Reply 3: It is our miss. The “TPR” means temperature, pulse and respiration. The full name has been added in this version.
Comment 4: I miss in Fig. 1 a communication layer which includes ZigBee, Bluetooth, NFC, WiFi, 5G. etc.
Reply 4: In the system, we use RFID for identification of resident and caregivers, ZigBee as the indoor locating, and wifi and 4G/5G as the communication channel between the portable devices and the data center. Bluetooth and NFC is not used in this platform. The sentence to describe the usage of RFID and Zee-bee has been rewritten.
Comment 5: Results disaggregated by the variable "resident", "relative" or "care giver" should be included.
Reply 5: It is our miss. The results of questionnaire are classified into two types, one is for residents or their relative, while the other one is for the caregivers. The results of the residents and their relatives are considered as a group. Since the measuring TPR operations are almost the same, the residents will not the influenced by the changed. But the relatives can quickly and easily know the health status of the residents. The results of questionnaires for the caregivers are missed in the last version and is added in this version.
Comment 6: When describing data from the residents, information regarding their current health status would be welcome (very good, good, regular or bad).
Reply 6: Since the system is implemented in daycare center, most of the residents are in subhealthy condition. Most of the elders are walkable but slow and have not-bad health status. Some residents have serious health conditions, like paralysis since of stroke, dementia, etc.; some are solitary elders but needs daycare. Since the elders will not use the ICT devices and system, the paper did not mention the health status of the elders in the previous version. For clearly describing the impact of the introduction of the system, we describe the health status of the residents in this revised version.
Comment 7: Some data relative to the care givers should be mentioned.
Reply 7: The results of questionnaires for the caregivers are missed in the last version and is added in this version.
Comment 8: The limitations of the study should be added.
Reply 8: Thanks for the reminds about the research limitation. The limitations of the study include the previous healthcare system usage of the institutions, the educational level of the relatives and/or the elders, which will influence the results of questionnaire. That is, the higher automatic degree the previous healthcare system, the higher adoption degree the users have; similarly, the higher educational level of the relatives and/or the elders, the higher satisfaction degree the relatives and the elders. We have added a discussion section to describe the limitation of the study in this version.
Comment 9: The whole manuscript should be revised by a professional proofreading service.
Reply 9: We have asked the edited service suggested by the Assistant Editor. Since the revision schedule is hurry, the proofreading will be provided later.
Round 2
Reviewer 1 Report
The literature sbould be strengthened.
Author Response
Q: The literature should be strengthened.
A: Thanks for the suggestions. In this version, we review the literature and extend their descriptions for several important literatures to let the readers can perceive the progress of service quality research in the academic. The following is some the paragraphs added in this revised version:
… They believed that quality was of the highest priority regarding the expectations for services of customers. PZB model [22] proposed insights and propositions from consumer’s viewpoint, revealing 10 dimensions (or indicators), namely, reliability, responsiveness, competence, access, courtesy, communication, credibility, security, understanding, and tangibles, that influence the consumers’ expectations about the service. …
… Wang, et.al. [23] collected contributed papers published from 1998 to 2013, that used “SERVQUAL (Service Quality)” model to gauge service quality, and gave a comprehensive survey about the trend and contribution of the service quality. In [28], Ko, et. al. also used SERVQUAL as well as importance performance analysis to identify the dissatisfactory factors in nursing home. The literature shows the service quality should be well gauged to find the potential problems. ….
… In [28], Ko, et. al. also used SERVQUAL as well as importance performance analysis to identify the dissatisfactory factors in nursing home. The literature shows the service quality should be well gauged to find the potential problems. …
…. Lim and Tang [26] utilized the market research techniques, i.e., SERVQUAL, to determine the expectation and perception of patients, and finally suggest the improvements, including tangibility, reliability, responsiveness, assurance, empathy and accessibility and affordability, to the hospital….